# TLR2 Signaling Pathway Combats *Streptococcus uberis* Infection by Inducing Mitochondrial Reactive Oxygen Species Production

**DOI:** 10.3390/cells9020494

**Published:** 2020-02-21

**Authors:** Bin Li, Zhixin Wan, Zhenglei Wang, Jiakun Zuo, Yuanyuan Xu, Xiangan Han, Vanhnaseng Phouthapane, Jinfeng Miao

**Affiliations:** 1MOE Joint International Research Laboratory of Animal Health and Food Safty, Key Laboratory of Animal Physiology and Biochemistry, College of Veterinary Medicine, Nanjing Agricultural University, Nanjing 210095, China; Lib_593114@163.com (B.L.); wzxsicau@163.com (Z.W.); 2019207011@njau.edu.cn (Z.W.); jiakunzuo@foxmail.com (J.Z.); xuyuanyuan@njau.edu.cn (Y.X.); 2Shanghai Veterinary Research Institute, Chinese Academy of Agricultural Sciences, Shanghai 200241, China; hanxgan@163.com; 3Biotechnology and Ecology Institute, Ministry of Science and Technology (MOST), Vientiane 22797, Lao PDR; wennaxianglaos@126.com

**Keywords:** *Streptococcus uberis*, mastitis, TLRs signaling pathway, mROS, inflammation

## Abstract

Mastitis caused by *Streptococcus uberis* (*S. uberis*) is a common and difficult-to-cure clinical disease in dairy cows. In this study, the role of Toll-like receptors (TLRs) and TLR-mediated signaling pathways in mastitis caused by *S. uberis* was investigated using mouse models and mammary epithelial cells (MECs). We used *S. uberis* to infect mammary glands of wild type, TLR2^−/−^ and TLR4^−/−^ mice and quantified the adaptor molecules in TLR signaling pathways, proinflammatory cytokines, tissue damage, and bacterial count. When compared with TLR4 deficiency, TLR2 deficiency induced more severe pathological changes through myeloid differentiation primary response 88 (MyD88)-mediated signaling pathways during *S. uberis* infection. In MECs, TLR2 detected *S. uberis* infection and induced mitochondrial reactive oxygen species (mROS) to assist host in controlling the secretion of inflammatory factors and the elimination of intracellular *S. uberis*. Our results demonstrated that TLR2-mediated mROS has a significant effect on *S. uberis*-induced host defense responses in mammary glands as well as in MECs.

## 1. Introduction

Mastitis is an inflammation caused by intra-mammary infection, leading to losses in the dairy industry [1]. *Streptococcus uberis* (*S. uberis*) is an environmental pathogen emerging as the most important mastitis-causing agent in some regions [2]. To make matters worse, there is growing evidence that it can infect people and pose a potential threat to human health [3]. Previous studies in our laboratory have demonstrated that persistent inflammation, including swelling, secretory epithelial cell degeneration, and polymorphonuclear neutrophilic leukocyte (PMN) infiltration, occurs in mammary tissue following injection with *S. uberis* [4]. Additionally, this inflammatory responce caused by *S. uberis* is lighter than that of caused by *E. coli* [4]. Additionally, these pathological responses are connected with *S. uberis* intracellular infection as it escapes the elimination of immune cells and induces persistent infection.

Activating pattern recognition receptors (PRRs) to produce natural inflammatory immune responses is important to control the intracellular infection induced by bacteria like *S. uberis* [5,6]. Toll-like receptors (TLR) family plays a critical role in these processes. Once activated by microbes, the MyD88-dependent pathway triggers producing inflammatory cytokines through nuclear factor (NF)-κB and mitogen-activated protein kinases. It is also possible through a TIR-domain-containing adapter through TRIF-dependent pathway to cause inflammation and this pathway is associated with inducing IFNs and stimulating T cell responses [7]. Previous research has found that PRRs are not only expressed by immune cells, but also by conventional non-immune cells, such as endothelial and epithelial cells, which also contribute to immune regulation [8].

Strandberg et al. first demonstrated that TLRs and their downstream molecules are expressed on bovine mammary epithelial cells (MECs) [8]. Ibeagha-Awemu et al. further revealed that the expressions of TLR4, MyD88, NF-κB, TIR domain-containing adapter molecule 2 (TICAM2), and IFN-regulatory factor 3 increase in bovine MECs were challenged by lipopolysaccharides [9]. These studies indicate that MECs could have a pivotal role in host defense with TLRs as their huge number in the mammary gland. Our laboratory has done a lot of research on the function of TLRs and MECs against *S. uberis* infection in vivo and in vitro models [4,10,11,12]. We find that TLRs, mainly TLR2 but not excluding TLR4, initiates a complex signaling network characterized by NF-κB and nuclear factor in activated T cells. In addition, it activates the secretion of cytokines and chemokines accompanied with its self-regulation pathways in response to *S. uberis* challenge [4].

Reactive oxygen species (ROS) is a kind of free radical including oxygen atoms, hydrogen peroxide (H_2_O_2_), superoxide anion (O^2−^), and hydroxyl radical (OH^−^) [13]. They are produced intracellularly through multiple mechanisms depending on the types of cell and tissue. However, the major ROS sources in mammalian cells are NADPH oxidase-induced ROS and mitochondrial-derived ROS (mROS) [12]. In most tissues, mROS from the respiratory chain is essential [14] because in innate immunity, mitochondria primarily fight bacterial infections through mROS, and this is evidenced by the fact that mROS modulates multiple signaling pathways including NF-B, C-Jun *N*-terminal kinase, and the caspase-1 inflammasomes [15]. Previous studies have shown that restricting pathogen-induced mROS impairs NF-κB activation, suggesting that mROS positively controls the NF-κB signaling pathway [16,17]. In addition, the production of mROS in immune cells like macrophages involves recruiting tumor necrosis factor (TNF) and receptor-associated factor (TRAF)6 to mitochondria and they also act as adaptors of the TLR signaling pathway [18]. Recently, MECs, the main cells for lactation in mammary tissue, have also been found to play a non-negligible role in the regulation of infection. Pathogens invading mammary tissue and epithelial cells can stimulate MECs to produce proinflammatory cytokines, anti-inflammatory factors, and chemokines such as TNF-α, interleukin (IL)-1β, IL-4, IL-6, IL-8, and IL-10 [19]. It is possible that MECs are involved in the generation of ROS in infections. However, few studies have investigated the interaction between TLRs and mROS against *S. uberis* infection in vivo and in vitro. Therefore, we defined whether TLR-induced mROS plays an important role against *S. uberis* infection in host and MECs.

## 2. Materials and Methods

### 2.1. Bacterial Strain, Cell Culture, and Treatment

*S. uberis* 0140J (American Type Culture Collection, Manassas, VA, USA) was inoculated into Todd–Hewitt broth (THB) supplemented with 2% fetal bovine serum (FBS; Gibco, New York, NY, USA) at 37 °C in an orbital shaker to mid-log phase (OD_600_ 0.4–0.6). MECs (American Type Culture Collection, Rockefeller, MD, USA) were incubated in Dulbecco’s modified Eagle’s medium (DMEM) with 10% FBS and plated at 80% confluence in 6-well cell culture cluster. After culture in serum-free DMEM for 4 h, the monolayer was treated with 40 nM NG25 (inhibitor of TGFβ-activated kinase 1; TAK1: Invitrogen, Carlsbad, CA, USA) for 24 h; 4 µm MK2206 (inhibitor of NADPHase: SellecK Chemicals, Houston, TX, USA) for 24 h; or transfected with 50 nM siTLR2 or/and siTLR4 for 72 h. SiECSIT with 20 nM were performed for 48 h using Lipofectamine 3000 reagent (Invitrogen). Transfection reagents and siRNA (siTLR2, siTLR4, siECSIT) were purchased from Guangzhou Ruibo Biotechnology Co., Ltd., Guangzhou, Guangdong, China. The sequences of siRNA were designed and listed as follows. siTLR2: GTCCAGCAGAATCAATACA; siTLR4: CAATCTGACGAACCTAGTA; siECSIT: GGTTCACCCGATTCAAGAA. Interference of TLR2, TLR4 (Appendix A) and ECSIT (Appendix A) gene identified by western blotting, which can be used to induce mastitis in cell models. The treated cells were infected with *S. uberis* at a multiplicity of infection (MOI) of 10 for 2 or 3 h at 37 °C. The supernatant and cells were collected separately and stored at −80 °C until use.

### 2.2. Mice and Treatment

Specific pathogen-free (SPF) clean-grade mice, including wild-type C57BL/6 (WT-B6), wild-type C57BL/10 (WT-B10), TLR2^−/−^ (C57BL/6), and TLR4^−/−^ (C57BL/10), aged 6–8 weeks (20 in total, each group includes two never-pregnant females, and three males) were purchased from Nanjing Biomedical Research Institute of Nanjing University (Nanjing, China) and bred under specific pathogen-free conditions in the Nanjing Agricultural University Laboratory Animal Center. Detailed description about the source for the TLR2^−/−^ and TLR4^−/−^ mice can be found on the website https://www.jax.org/strain/004650 and https://www.jax.org/strain/007227 respectively. These healthy pregnant mice were housed in individual cages and provided water and food ad libitum. All experimental protocols were approved by the Regional Animal Ethics Committee and were in compliance with animal welfare act regulations as well as the guide for the care and use of laboratory animals. Our protocol number approved by the Animal welfare committee is N1418044. After 1 week of adaptive feeding, the female and male mice were kept in cages in a ratio of 2:1, so that they were mated and conceived.

After parturition for 72h, all experimental groups of female mice were infused with 100 colony-forming units (CFU) *S. uberis* in volume of 50 μL into the left 4th (L4) and right 4th (R4) teats. The offspring were weaned 2 h prior to experimental infusion. Following administration of ether anesthesia, the L4 and R4 teats were moistened with 75% ethanol, a 33-gauge needle fitted to a 1 mL syringe was gently inserted into the mammary duct, and 50 μL of *S. uberis* was slowly infused. At 24 h post *S. uberis*-infusion (PI), all mice were euthanized, and the mammary gland was aseptically collected and stored at −80 °C until analyzed.

The mammary gland was fixed in 10% neutral buffered formalin. Sections of 5 μm thickness were stained with hematoxylin and eosin. Mammary gland tissues were weighed and homogenized with sterile phosphate buffered saline (PBS) (1:5, *w/v*) on ice. After centrifuged at 500× *g* at 4 °C for 40 min, the supernatant was centrifuged again. The second supernatant was collected and stored at −80 °C until assayed.

### 2.3. Histological Observation and Immunohistochemistry

The mammary tissue fixed in 10% neutral buffered formalin was trimmed and flushed in water for at least for 4 h, and then dehydrated in alcohol solutions ranging from 75% to 100%, with 5% increase at 1 h intervals. After soaking in xylene, the tissues were embedded in wax for 3 h at 60 °C. Slices (5 μm thick) were cut and stained with hematoxylin and eosin. The histological changes, including PMN infiltration, bleeding and degeneration, and adipose tissue loss, were analyzed by light microscopy (BH2; Olympus, Tokyo, Japan) at a magnification of 40×. Specifically, Leukocyte infiltration, mainly lymphocytes and PMN, was categorized in four tissue areas: (1) teat cistern lining; (2) gland cistern lining; (3) gland cistern parenchyma; and (4) deep parenchyma. Prevalence of these cells was estimated for each tissue section at 250× and assigned a score of 1, 2, or 3 where 1: none to few leukocytes present; 2: moderate leukocyte infiltration; and 3: marked leukocyte infiltration. Results were presented as average leukocyte infiltration score for each section of tissue characterized. Bleeding and degeneration in tissue samples were characterized using a score where 1: none to few bleeding and degeneration; 2: moderate bleeding and degeneration; and 3: marked bleeding and degeneration. Results are expressed as the mean bleeding and degeneration of each tissue section characterized. Area occupied by adipose tissue in secretory parenchyma tissue samples was estimated using a score where 1: less than 20% adipose tissue; 2: 20% to 50% adipose tissue; and 3: more than 50% adipose tissue. Alveolar lumen characterization and adipose tissue estimation were expressed as frequency percentages of each assigned score. The histological changes, including PMN infiltration, bleeding and degeneration, and adipose tissue loss, were observed by light microscopy (BH2; Olympus, Tokyo, Japan) at a magnification of 40× and then were analyzed by double bland method [20].

Immunohistochemical staining was performed as follows. Tissue sections were washed with PBS, then covered with 3% H_2_O_2_ for 15 min at 37 °C to inhibit further endogenous peroxidase activity. Tissue slices were blocked with 5% bovine serum albumin and incubated with antibodies against MyD88, TRAF6, ECSIT, and TRIF (Cell Signaling Technology, Danvers, MA, USA), at 4 °C in a humidified chamber. Overnight, biotinylated anti-rabbit IgG (Boster Bio-Technology, Wuhan, China) was incubated for 30 min at 37 °C. After rehydration, the sections were incubated with avidin–biotin peroxidase complex for 40 min at 37 °C. Finally, the sections were washed and bound conjugates were revealed by diaminobenzidine staining (Boster Bio-Technology). The sections were imaged with microscope and characterized quantitatively by using the Image Pro-Plus 5.0 image-analysis software (Media Cybernetics, Silver Spring, MD, USA). Integrated optical density (IOD) indicates the total amount of staining material in each section.

### 2.4. RNA Extraction and Quantitative Real-Time Polymerase Chain Reaction (qPCR)

PCR was carried out as previously described [10]. Total RNA was extracted by TRIzol reagent (TaKaRa, Dalian, China). Corresponding cDNA was obtained using reverse transcriptase and Oligo (dT) 18 primer (TaKaRa). An aliquot of the cDNA was mixed with 25 µL SYBR^®^ Green PCR Master Mix (TaKaRa) and 10 pmol of each specific forward and reverse primer. All mixed systems were analyzed in an ABI Prism 7300 Sequence Detection System (Applied Biosystems, Waltham, MA, USA). Fold changes were calculated as 2^−ΔΔCt^. All primer sequences (Appendix A) were synthesized by Invitrogen Biological Company (Shanghai, China).

### 2.5. Total Protein Extraction and Western Blotting

Intracellular protein levels were determined by Western blotting analysis. GAPDH (Bioworld, USA) was employed to ensure equal loading. Cells were washed twice in ice-cold PBS, lysed with RIPA buffer (Beyotime, Nantong, China) added protease inhibitor PMSF (Beyotime, Nantong, China) by incubating on ice for 30 min in an Eppendorf tube. The supernatants were collected by centrifuging at 5000× *g* for 10 min at 4 °C protein concentration was determined by bicinchoninic acid assay (BCA) (Bebytime, Nantong, China). Then, 10% gel was used, and 10 μL of protein sample was added per hole. Extracts with equal amounts of proteins were solubilized by SDS sample buffer (BioRad, Califonia, USA), separated by SDS-PAGE, and transferred to polyvinylidene difluoride membranes (Millipore, Bedford, MA, USA). The membranes were blocked with 5% non-fat milk diluted in Tris buffered saline with Tween-20 (TBST) for 2 h at room temperature, and hybridized overnight with primary antibody at 4 °C Primary antibodies were diluted as follows: GAPDH (1:10,000), MyD88 (1:1000), TRAF6 (1:1000), ECSIT (1:1000), TRIF (1:1000). Before and after incubation with the secondary antibodies at room temperature for 2 h, the membranes were washed 3 times with TBST. Secondary antibody is horseradish peroxidase (HRP)-linked anti-rabbit IgG (CST, Massachusetts, USA, 1:10,000). The signals were detected by an ECL western blot analysis system (Tanon, Shanghai, China). Analysis of bands was quantified with Image J software (NIH, Bethesda, MD, USA).

### 2.6. Measurement of Reactive Oxygen Species (ROS) and Mitochondrial Reactive Oxygen Species (mROS)

Intracellular ROS was evaluated by staining MECs with dichloro-dihydrofluorescein diacetate (DCFH-DA) (Beyotime, Nantong, China), a fluorescent ROS-sensitive indicator that freely permeates cell membranes. mROS was assessed by MitoSOX (Thermo, Waltham, MA, USA), a fluorescent mROS-sensitive indicator. Briefly, after incubating with 10µM DCFH-DA for 30 min at 37 °C or 5 μM MitoSOX for 20 min, cells were washed 3 times in phosphate buffered saline (PBS) and detached. The cells were centrifuged at 400× *g* for 5 min, resuspended in PBS, and immediately analyzed by flow cytometry using FACSCanto (BD, Franklin, NJ, USA). Ten thousand cells per sample were analyzed using CellQuest Pro acquisition and Flow Jo software [21].

### 2.7. Assay of TNF-α, IL-1β, and IL-6 by ELISA

The levels of TNF-α, IL-1β, and IL-6 in mammary glands and MECs were measured by ELISA (Rigor Bioscience, Beijing, China). Prepared standards (50 μL), and antibodies (40 μL) labeled with enzyme (10 μL) were reacted for 60 min at 37 °C and the plate was washed five times. Chromogen solutions A (50 μL) and B (50 μL) were added and incubated for 10 min at 37 °C. Stop solution (50 μL) was added and optical density value was measured at 450 nm within 10 min. Qualitative differences or similarities between the control and experimental groups were consistent throughout the study.

### 2.8. Detection of NAGase, Total Antioxidant Capacity (T-AOC), Superoxide Dismutase (SOD), Malondialdehyde (MDA), and Uncoupling Protein 2 (UCP2)

The activities or levels of NAGase, total antioxidant capacity (T-AOC), superoxide dismutase (SOD), malondialdehyde (MDA), and uncoupling protein 2 (UCP2) were determined using commercial kits purchased from Nanjing Jiancheng Bioengineering Institute (China).

### 2.9. Viable Bacterial Count Assay

Viable bacteria were enumerated as colony-forming units (CFU) on THB agar. The mammary glands were aseptically homogenized with sterile PBS (1:5, *w/v*). The supernatants were spread on plates. CFUs were counted by the spread plate method after incubation for 12 h at 37 °C.

MECs and MECs with siECSIT were incubated in DMEM with 10% FBS and plated at 80% confluence in 6-well plates. After culture in serum-free DMEM for 4 h, at mid-exponential phase (OD_600_ 0.4–0.6), *S. uberis*-infected cells were washed 3 times with PBS containing 100 mg/mL gentamicin, followed by gentamicin-free PBS. Cells were pelleted at 1.4× *g* for 10 min. The same number of cells were lysed with sterile triple distilled water, and CFUs were counted by the spread plate method after incubation for 12 h at 37 °C [22].

### 2.10. Statistical Analysis

Results were analyzed using GraphPad Prism 5.0 software (GraphPad Software Inc., La Jolla, CA, USA). Data were expressed as means standard error of the mean (SEM). Differences were evaluated by one-way analysis of variance followed by post-hoc tests. Significant differences were considered at *p* < 0.05.

## 3. Results

### 3.1. TLR2 Mediates Tissue Damage and Anti-S. uberis Infection in Mammary Glands

*S.uberis* belongs to gram-positive bacteria which is mainly recognized by TLR2. However, previous research has demonstrated that the role of TLR4 could not be ignored in *S. uberis* infection due to the close relationship, similar structure, and function between TLR2 and TLR4 [3,23,24]. In this work, we explored the roles of TLR2 and TLR4 in *S. uberis* infections in TLR2^−/−^ and TLR4^−/−^ mice to further understand the molecular defense mechanism in *S. uberis* mastitis. No histological changes were observed in WT-B6 or WT-B10 mammary glands of control mice, whereas, there was some suspicion of tissue damage in TLR2^−/−^ and TLR4^−/−^ control mice (Figure 1A). Inflammation and tissue damage appeared in mammary tissue after infection with *S. uberis* in all challenged groups. These responses were characterized by PMN infiltration, increased bleeding and epithelial cell degeneration, and excess adipose tissue. Compared with WT-B6 mice, TLR2 deficiency induced more severe pathological damage. A higher score was presented for the three indexes mentioned above and there were significant increases in bleeding and degeneration, as well as excess adipose tissue (*p* < 0.05; Figure 1B). However, TLR4 deficiency caused insignificant inflammation and tissue damage during *S. uberis* challenge comparing WT-B10 mice (Figure 1A,C).

*N*-acetyl-β-d-glucosaminidase (NAGase), a marker enzyme of MECs and mammary gland damage, was significantly elevated in TLR2^−/−^ mice, but not in TLR4^−/−^ mice when compared with WT mice at 24 h post-challenge (*p* < 0.05; Figure 1D and Appendix A). Similarly, the number of bacteria in the mammary tissue of TLR2^−/−^ mice was higher than that of in WT-B6 mice (*p* < 0.05), but there was no significant difference between TLR4^−/−^ and WT-B10 mice. We conclude that it is TLR2 that primarily mediated the tissue damage and anti-bacterial effect in mammary glands during *S. uberis* infection.

### 3.2. TLR2 and TLR4 Deficiencies Affect the Secretion of Cytokines in S.uberis Infection

Previously, it has been reported that the secretion of proinflammatory cytokines in mammary glands can more accurately reflect the level of inflammation [17,25]. Although inflammation contributes to fighting infection, massive release of cytokines can cause irreversible damage to tissues. Here, we investigated the level of TNF-α, IL-1β and IL-6 in response to *S. uberis* infection in TLR2^−/−^ and TLR4^−/−^ mice (Figure 2A,B). *S. uberis* challenge significantly increased TNF-α level in WT, TLR2^−/−^ and TLR4^−/−^ mice (*p* < 0.05). Compared with corresponding control mice, TNF-α and IL-1β in TLR2^−/−^ mice and TNF-α in TLR4^−/−^ mice significantly decreased (*p* < 0.05). These results indicated that TLR2 and TLR4 deficiencies affected the secretion of cytokines in *S.uberis* infection.

### 3.3. MyD88-Dependent Pathway Predominates in S. uberis Infection

After TLR activation, MyD88 dependent and independent pathways are critical to host responses [26]. TLR2 and TLR4 can activate MyD88-dependent signaling pathway to produce cytokines and TLR3 and TLR4 can activate TRIF-dependent signaling, which activates NF-kB and IRF3 resulting in the induction of proinflammatory cytokine genes and type I IFNs [27]. We assessed the expressions of MyD88 and TRIF, respectively, by immunohistochemistry in mammary glands, as they are the key molecules in the MyD88-dependent or independent signaling pathways. There was a significant increase in the expression of MyD88 instead of TRIF in WT, TLR2^−/−^, and TLR4^−/−^ mice after challenged with *S. uberis* (*p* < 0.05). However, compared to WT mice, the knockout of TLR2 or TLR4 reduced the expression level of MyD88 during *S. uberis* infection. (*p* < 0.05; Figure 3A,B). MECs as the main functional cells in mammary glands, our previous study has established that they play a key role in anti-infection response in mammary glands. Furthermore, we detected that interference of TLR2 and/or TLR4 by specific siRNA significantly decreased MyD88 expression in *S. uberis* infection as well (*p* < 0.05; Figure 3C–E). These data suggested that following TLR activation, the MyD88-dependent pathway predominated during *S. uberis* infection in MECs and in mammary glands.

### 3.4. TRAF6 and ECSIT Participate in Signal Sensing from Toll-Like Receptors (TLRs) in S. uberis Infection

We next evaluated the expression levels of TRAF6 and evolutionarily conserved signaling intermediate in Toll pathways (ECSIT) which are downstream targets of the MyD88 signaling pathway, using immunohistochemistry in mammary glands of mice. The TRAF6 adaptors increased dramatically in all mice after *S. uberis* infection (*p* < 0.05), although TLR2 or TLR4 deletion weakened the expression of ECSIT compared with WT mice during *S. uberis* challenge (*p* < 0.05; Figure 4A,B). Similarly, in MECs, interfering TLR2 or TLR4 significantly reduced the expressions of TRAF6 and ECSIT after *S. uberis* infection (*p* < 0.05; Figure 4C–E). Interestingly, there is no significant difference between TLR2^−/−^ and TLR4^−/−^ mice in the expressions of TRAF6 and ECSIT during *S. uberis* infection. These results confirmed that TRAF6 and ECSIT downstream of the MYD88 pathway mediated the anti-*S. uberis* response in mice and in MECs.

### 3.5. TLRs Mediate Redox Status in Mammary Glands During S. uberis Infection

TRAF6 activated by TLRs transfers from cytoplasm to mitochondria, where it engages ECSIT to produce mROS inducing cellular anti-bacterial responses [28]. Since the level of ROS in tissue cannot be detected well, we analyzed total antioxidant capacity (T-AOC), superoxide dismutase (SOD), malondialdehyde (MDA), and uncoupling protein 2 (UCP2) in mammary glands to indirectly reflect the antioxidant levels. The levels of MDA and UCP2 were significantly increased due to the infection of *S. uberis* in WT, TLR2^−/−^, and TLR4^−/−^ mice (*p* < 0.05; Figure 5A,B). Although there was no obvious distinction between MDA (*p* > 0.05), the absence of TLR2 further reduced UCP2 (*p* < 0.05). The level of T-AOC was significantly lower in all groups after *S. uberis* challenge compared with control accordingly (*p* < 0.05). Deletion of TLR4 rather than TLR2 significantly decreased SOD activity after *S. uberis* infections (*p* < 0.05). These results indicated that the host’s oxidation level did change after *S. uberis* infection and these changes were related to the TLRs signaling pathway.

We aimed to clarify whether MECs were involved in the change in redox status and had a crucial role in *S. uberis* infection after activating TLR signaling pathway. Therefore, we interfered with the expression of TLR2 and/or TLR4 in MECs and detected ROS, mROS and UCP2 levels. *S. uberis* infection caused a significant increase in the levels of ROS and mROS (Figure 5C,D), but interfering with TLR2 significantly reduced ROS and mROS levels (*p* < 0.05). SiTLR4 also decreased their levels to some extent, but no significant difference was observed (*p* > 0.05). The expression of UCP2 decreased but interfering with TLR2 reversed this change (*p* < 0.05). Taken together, these results demonstrated that infection with *S. uberis* changed the redox status in mammary glands as well as in MECs, and TLR2 played an essential role in this process, especially in MECs.

### 3.6. mROS Plays an Important Role Against S. uberis Infection in Mammary Epithelial Cells (MECs)

GKT137831, a specific inhibitor of NADPH oxidase 1 (NOX1) and NOX4, and NG25, an inhibitor of TAK1 [29,30], were used to suppress ROS generation from NOX complexes and down-regulate the production of proinflammatory cytokines respectively. GKT137831 and NG25, simultaneously or separately, reduced the generation of ROS after challenge with *S. uberis* but cannot change the level of mROS (*p* < 0.05; Figure 6A). The bacterial counts of *S. uberis* in MECs were significantly higher in the inhibitor-treated groups (*p* < 0.05) (Figure 6B). In addition, we measured the levels of cytokines and obtained similar results (Appendix A). We inhibited the production of mROS by siECSIT to establish mROS role in regulating inflammation and anti-*S. uberis* activity. ROS and mROS levels decreased significantly after using siECSIT (*p* < 0.05; Figure 6C). Similar results were observed at TNF-α, IL-1β, and IL-6 expressions: their levels were up-regulated after *S. uberis* infection but siECSIT can reduce their expressions (Figure 6D). Meanwhile, the bacterial counts of *S. uberis* in MECs were significantly higher in the siECSIT treatment group (Figure 6E). These results demonstrated that mROS does play an important role against *S. uberis* infection in MECs.

## 4. Discussion

The intrusion signal (from molecules broadly shared by pathogens that could be recognized by immune system) of intracellular bacteria captured by PRRs is crucial for host to control inflammation and pathogen proliferation [31]. TLRs are one of the most ancient, conserved components of the immune system, and it has been established by our laboratory that they can sense and respond to *S. uberis* [4]. *S. uberis* is a kind of Gram-positive bacterium and TLR2 is the principal receptor that can sense its invasion [32]. However, TLR2 and TLR4 share the same delivery system, and current studies have not yet distinguished their exact roles in defending *S. uberis* infection. We used TLR2^−/−^ and TLR4^−/−^ mice to investigate the roles of these two high-correlation receptors in *S. uberis* infections thoroughly for the first time. Deficiency of TLR2, rather than TLR4, induced a more severe inflammatory response and tissues damage in mammary gland and bacterial viability was higher accordingly. These results confirmed that TLR2 detected *S. uberis* infection initiated the antibacterial immunological reaction and controlled the inflammatory status in mammary glands.

Proinflammatory cytokines, such as TNF-α, IL-1β, and IL-6, are secreted following activating TLRs and their respective downstream signaling pathways, mainly in immune cells [33]. They are involved in the upregulation of inflammatory reactions and play a role in regulating host defense response against pathogens to mediate innate immune responses. In this study, TNF-α, the initial factor in the cytokine storm, increased dramatically after *S. uberis* challenge in all variant mice. However, a similar change was only seen in TLR2^−/−^ mice for IL-1β. No significant change was observed for IL-6. These findings were consistent with previous reports that the expressions of TNF-α, IL-1β and IL-6 have a chronological order. For the samples detected here, they were only expressed at 24 h post-infection [34]. Compared with WT mice, the levels of TNF-α and IL-1β were obviously decreased in TLR2^−/−^ mice during *S. uberis* infection. This further demonstrated the important role of TLR2 in the interaction between *S. uberis* infection and the host. Similar changes were also seen in TNF-α levels in TLR4 ^−/−^ mice, which may be due to the complexity of the inflammatory response network after infection. Both positive and negative inflammatory factor feedbacks were present in *S. uberis*-infected mammary glands. The secondary inflammation induced by initial inflammatory factors may be caused in part by activation of TLR4. Therefore, the down-regulation of TNF-α caused by TLR4 deficiency was different from the loss of TLR2 expression, which cannot neutralize the inflammatory response caused by *S. uberis*.

Two distinct signaling pathways, the MyD88-dependent and TRIF-dependent pathways, are triggered by dimerized and activated TLRs [35]. Our experiments in vivo found that MyD88, instead of TRIF, was affected significantly in *S. uberis* infection, and thus confirmed that the MyD88-dependent pathway predominated in this process. This phenomenon also exists in other bacterial infections. For example, Wiersinga et al. reported similar results in *Burkholderia pseudomallei* infection [36]. In the MyD88-dependent pathway, MyD88 recruits IL-1 receptor-associated kinases and then phosphorylates and activates TRAF6 which in turn polyubiquinates TAK1, and induces the secretion of inflammatory cytokines during *S. uberis* infection in vivo and in vitro [4,37]. Recently, it has been suggested that activated TRAF6 translocates to the mitochondria, leading to ECSIT ubiquitination, and resulting in increased mROS generation [38]. This signaling pathway plays an important role in innate immune responses against intracellular bacteria. A recent study also showed that macrophages depleted of ECSIT and TRAF6 reduce TLR-induced ROS levels and significantly impair their ability to kill intracellular bacteria [26]. Sonoda et al. similarly found that estrogen-related receptor α and PPAR gamma Coactivator-1 β (PGC-1β) act together as key effectors of IFN-γ-induced mitochondrial ROS production and host defense [39]. Our study emphasized the importance of mROS in killing bacteria. Since detecting ROS in tissues is difficult, previous research has always detected the presence of components of the antioxidant system in organs, such as T-AOC, SOD, MDA, and UCP2 to indirectly reflect the production of ROS [25,40]. Our results showed that *S. uberis* challenge caused significant changes in redox status in mammary glands. This indicates that the TLRs/MyD88/TRAF6/ECSIT/mROS axis participated in the defense responses to *S. uberis* infection.

The inflammatory phenomena in mammary glands involve integrated responses of all kinds of mammary cells including macrophages, PMNs, lymphocytes, MECs, and even matrix cells [41]. In the past decade, we have paid more attention to the defensive ability of MECs because they are the most numerous cells in the udder, and we have detected TLRs-mediated signaling pathways and secretion of more than 40 cytokines in MECs [33]. In addition, we showed that *S. uberis* adhered and internalized in MECs indicating that MECs are one of the main target cells of *S. uberis* (data not published). Intriguingly, MECs are not real immune cells and have distinctive responses to bacterial infections. For example, we have found that the PI3K/Akt/mTOR pathway in MECs generates a positive contribution to inflammation following viable *S. uberis* challenge, which is not consistent with the usual situation in some immune cells [11]. Thus, we treated MECs with specific siRNAs targeting to TLR2 and/or TLR4 and then evaluated the effect of *S. uberis* challenge on the expression of key adaptor proteins. The results confirmed that the MyD88-dependent pathway predominated in *S. uberis*-infected MECs after TLR2 activation. A similar signal transfer process is reported in macrophages infected with *Mycobacterium tuberculosis,* another Gram-positive bacterium [42]. In the present study, we were interested in whether the TLR2/MyD88/TRAF6/ECSIT axis regulated the production of ROS. Suppression of TLR2, but not TLR4, reduced the level of ROS and mROS in MECs after *S. uberis* challenge. This result was further confirmed by the detection of UCP2 that separates oxidative phosphorylation from ATP synthesis and thus improves the production of ROS and mROS [11]. It is interesting that UCP2 levels in MECs decreased while increased in tissues during *S. uberis* infection. This may be because ROS level increased with the *S. uberis* infection, and MECs are not professional immune cells; it cannot cope with this rapid oxidative change, so UCP2 increased at the same time due to a self-protection mechanism. The mammary gland has not only MECs, but also many endothelial cells and immune cells. The decline in UCP2 is the result of the combined force of various cells. Besides, the mammary glands indeed need stronger oxidizing power to resist infection by S. uberis. It also indicates that MECs have their own unique defense mechanisms and more research is required to prove this.

Initially, mROS was considered to be a by-product of bio-oxidation, whose synthesis cannot be regulated. Numerous studies have established that oxidative phosphorylation in mitochondria is the main pathway for mROS production and the main source of ROS [43]. Expressing catalase in mitochondria could effectively reduce the production of mROS, thereby reducing the killing effect of macrophages on pathogens, indicating that mROS is a key driver in the process of antibacterial activity [10]. Our previous study shows that TLR2 regulates the generation of ROS including mROS during *S. uberis* infection both in vivo and in vitro [4]. We suggested that TLR2-mediated mROS was involved in *S. uberis* infection. To confirm our hypothesis, GKT137831 and NG25 were used alone or simultaneously to suppress ROS from NOX complexes and the production of proinflammatory cytokines. We found that the antibacterial activity of MECs was restrained to some extent, and this established that ROS from NOX complexes and cytokines were involved in the host defensive reaction, which was consistent with our previous study [11]. Furthermore, we inhibited mROS synthesis by siECSIT and investigated the changes in inflammation and the effect of reducing mROS on bacterial viability. The bacterial counts of *S. uberis* were significantly higher in the siECSIT treatment group. These results demonstrated that TLR2-mediated mROS was a key factor against *S. uberis* infection in MECs. It is worth noting that because knockout of TLR4 also reduced ECSIT expression, we cannot eliminate that TLR4 may also mediate mROS production.

In conclusion, mROS participated in the host response against *S. uberis* infection, and TLR2 was involved in sensing *S. uberis* invasion and controlling mROS production by regulating the expressions of TRAF6 and ECSIT. Additionally, the function of mROS against *S. uberis* infection probably relied on its ability to regulate cytokine levels, thereby controlling the level of inflammation. However, we must point out that due to the limitations of the current experiment methods, we cannot provide more sufficient in vivo model data to further validate our conclusions. Notwithstanding this limitation, this study increased our understanding of the molecular defense mechanisms in S. uberis mastitis and provided theoretical support for the development of prophylactic strategies for this critical disease. This study increased our understanding of the molecular defense mechanisms in *S. uberis* mastitis and provided theoretical support for the development of prophylactic strategies for this critical disease.

## Figures and Tables

**Figure 1 cells-09-00494-f001:**
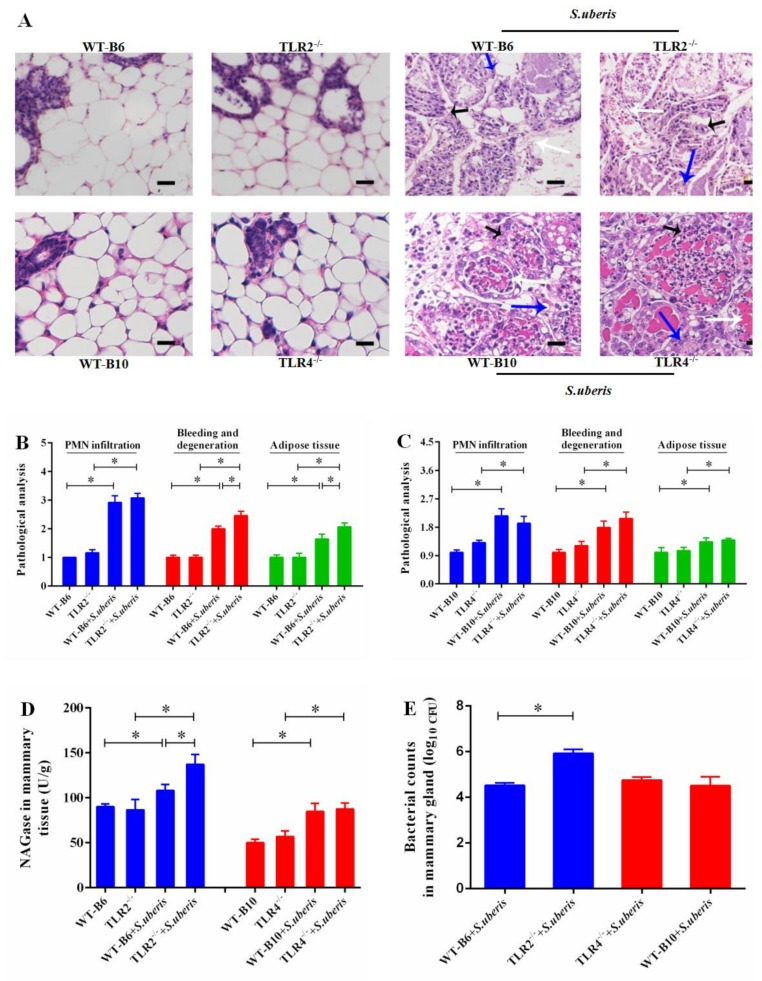
TLR2 mediates tissue damage and anti-*S. uberis* infection in mammary gland. (**A**) Mammary gland staining by hematoxylin and eosin of WTB6 (isotype-matched negative control of TLR2^−/−^), WTB10 (isotype-matched negative control of TLR4^−/−^), TLR2^−/−^, TLR4^−/−^ mice with or without *S. uberis*. Polymorphonuclear neutrophilic leukocyte (PMN) infiltration (black arrow), the bleeding and degeneration (white arrow), adipose tissue (blue arrow). Images are representative of *n* = 6 animals per genotype. Scale bars, 50 μm. (**B**,**C**) The bleeding and degeneration, PMN infiltration, and adipose tissue were observed by light microscopic and scored by double blind method. B stands the scores for TLR2^−/−^ group and C is for TLR4^−/−^ group (**D**) NAGase activity was analyzed in mammary gland. (**E**) Viable bacteria were counted via the plate with Todd–Hewitt broth (THB) agar medium. Experiments D and E were repeated three times. All data were presented as the means ± SEM (*n* = 6). * (*p* < 0.05) = significantly different between the indicated groups.

**Figure 2 cells-09-00494-f002:**
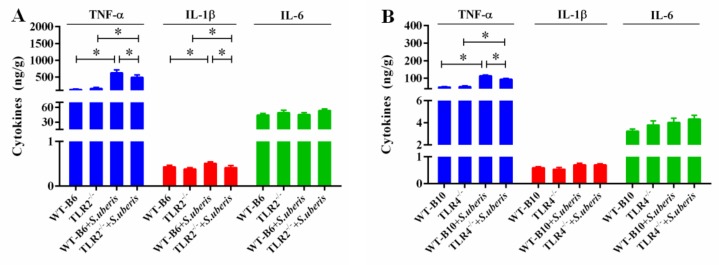
TLR2 and TLR4 deficiencies affect the secretion of cytokines in *S.uberis* infection. (**A**,**B**) The protein expressions of TNF-α, IL-1β and IL-6 were determined by ELISA in mammary gland of WTB6, WTB10, TLR2^−/−^ and TLR4^−/−^ mice with or without *S. uberis*. A is for the protein expressions for TLR2^−/−^ group and B is for the TLR4^−/−^ group. Experiments were repeated three times and all data were presented as the means ± SEM (*n* = 6). * (*p* < 0.05) = significantly different between the indicated groups.

**Figure 3 cells-09-00494-f003:**
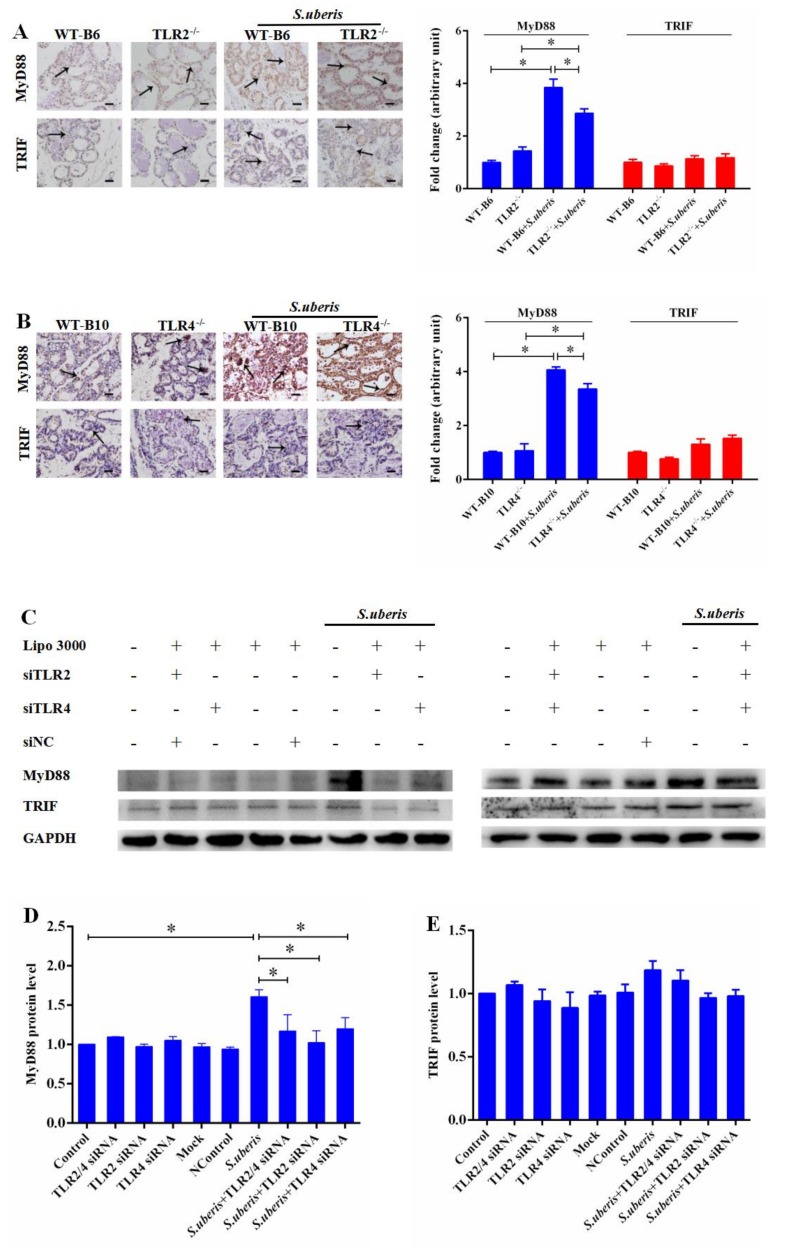
MyD88 dependent pathway predominates in *S. uberis* infection. (**A**) Immunohistochemistry was used to analyze the protein expression levels of MyD88 and TRIF in mammary gland of WTB6 and TLR2^−/−^ mice with or without *S. uberis*. (**B**) Immunohistochemistry was used to analyze the protein expression levels of MyD88 and TRIF in mammary gland of WTB10 and TLR4^−/−^ mice with or without *S. uberis*. Images are representative of *n* = 6 animals per genotype. Scale bars, 100 μm. Data were presented as the means ± SEM (*n* = 6). * (*p* < 0.05) = significantly different between the indicated groups. (**C**,**D**,**E**) The protein expression levels of MyD88 and TRIF of WTB6, WTB10, TLR2^−/−^, TLR4^−/−^ mice with or without *S. uberis* were determined by Western blot in mammary epithelial cells (MECs). For quantitative analysis, bands were evaluated densitometrically with Image J analyzer software normalized for GAPDH density. Experiments were repeated three times and data were presented as the means ± SEM (*n* = 3). * (*p* < 0.05) = significantly different between the indicated groups.

**Figure 4 cells-09-00494-f004:**
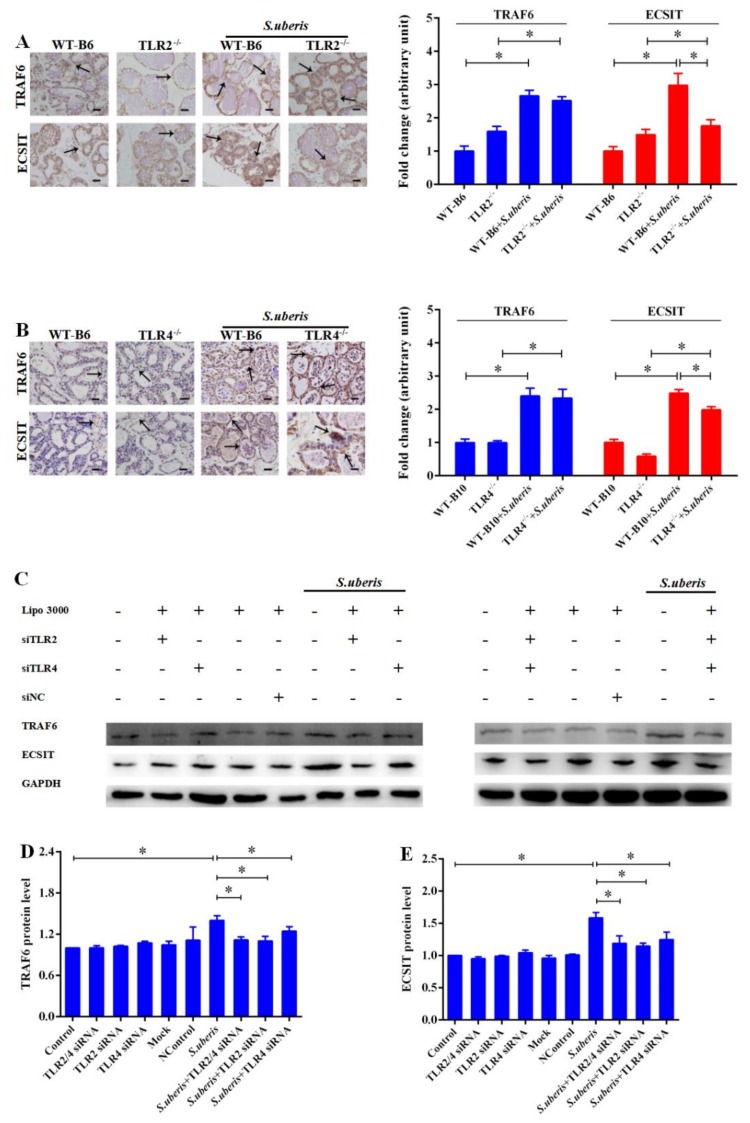
TRAF6 and ECSIT participate in sensing signal from Toll-like receptors (TLRs) in *S. uberis* infection. (**A**) Immunohistochemistry was used to analyze the protein expression levels of TRAF6 and ECSIT in mammary gland of WTB6 and TLR2^−/−^ mice with or without *S. uberis*. (**B**) Immunohistochemistry was used to analyze the protein expression levels of TRAF6 and ECSIT in mammary gland of WTB10 and TLR4^−/−^ mice with or without *S. uberis*. Images are representative of *n* = 6 animals per genotype. Scale bars, 100 μm. Data were presented as the means ± SEM (*n* = 6). * (*p*< 0.05) = significantly different between the indicated groups. (**C**,**D**,**E**) The protein expression levels of TRAF6 and ECSIT of WTB6, WTB10, TLR2^−/−^, TLR4^−/−^ mice with or without *S. uberis* were determined by Western blot in MECs. For quantitative analysis, bands were evaluated densitometrically with Image J analyzer software normalized for GAPDH density. Experiments were repeated three times and data were presented as the means ± SEM (*n* = 3). * (*p* < 0.05) = significantly different between the indicated groups.

**Figure 5 cells-09-00494-f005:**
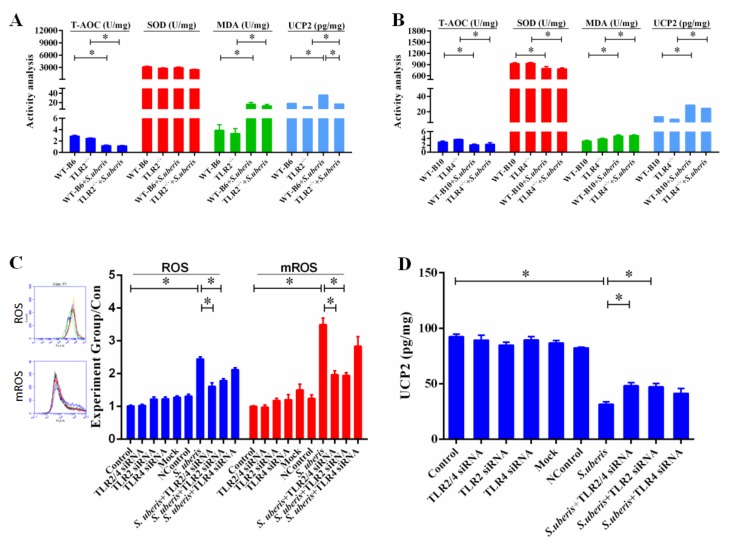
TLRs mediate redox status in mammary gland during *S. uberis* infection. (**A**,**B**) The protein expressions of total antioxidant capacity (T-AOC), superoxide dismutase (SOD), malondialdehyde (MDA), and uncoupling protein 2 (UCP2) were determined by kits in mammary gland of WTB6, WTB10, TLR2^−/−^, TLR4^−/−^ mice with or without *S. uberis*. A is for the protein expressions for TLR2^−/−^ group and B is for the TLR4^−/−^ group. Data are presented as the means ± SEM (*n* = 6). * (*p* < 0.05) = significantly different between the indicated groups. (**C**) CellQuest Pro acquisition and analysis software analyzed the levels of reactive oxygen species (ROS) and mitochondrial reactive oxygen species (mROS) in MECs of Control, siTLR2, siTLR4, and siTLR2/4 groups with or without *S. uberis*. (**D**) The activity of UCP2 was determined by ELISA in MECs of Control, siTLR2, siTLR4 and siTLR2/4 groups with or without *S. uberis*. Data were presented as the means ± SEM (*n* = 3). * (*p* < 0.05) = significantly different between the indicated groups. All experiments were repeated three times.

**Figure 6 cells-09-00494-f006:**
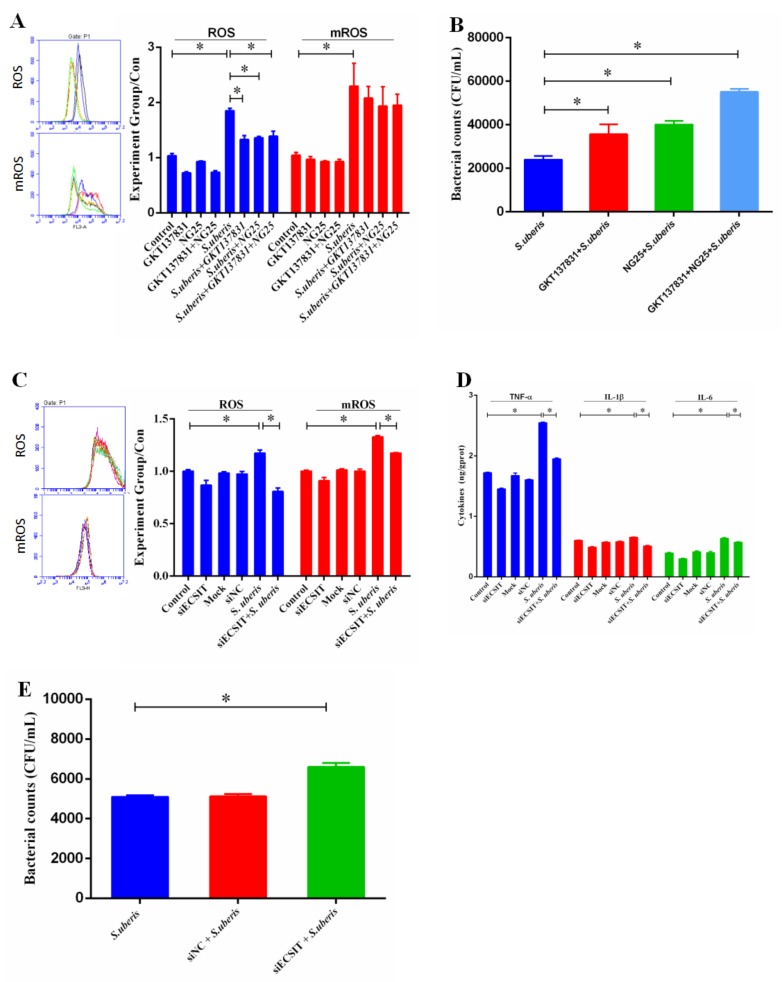
mROS plays an important role in anti-*S. uberis* infection in MECs. (**A**) The levels of ROS and mROS after using GKT137831 and NG25 simultaneously or separately with or without *S. uberis* infection in MECs. (**B**) Viable bacteria were counted via the plate with THB agar medium after using GKT137831 and NG25 simultaneously or separately during *S. uberis* infection in MECs. (**C**) The levels of ROS and mROS after using siECSIT in MECs of Control and siECSIT groups with or without *S. uberis* infection. (**D**) The expressions of TNF-α, IL-1β and IL-6 after using siECSIT in MECs of control and siECSIT groups with or without *S. uberis* infection. (**E**) Bacteria counts after using siECSIT in MECs of Control and siECSIT groups during *S. uberis* infection. All experiments were repeated three times and all data were presented as the means ± SEM (*n* = 3). * (*p* < 0.05) = significantly different between the indicated groups.

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
