# Peer review of "TLR2 Signaling Pathway Combats Streptococcus uberis Infection by Inducing Mitochondrial Reactive Oxygen Species Production"

_cells, 2020, doi:10.3390/cells9020494_

Round 1

Reviewer 1 Report

The submitted manuscript is well designed and scientifically sound. The results are interesting for readers. However, there are some comments which I hope this manuscript will be better. 

Comment1:

The study represents the TLR2 mediated signaling pathway during the S. uberis infection.

The author successfully showed that TLR2 mediates the signaling pathway is involved in the mROS generation in in vitro model (MEC cells). However, in the mouse model, the evidence of TLR2 mediated signaling is not adequate. Is there any method in which the author elucidates this phenomenon in in vivo model? If not, the author may need to discuss it.

Comment2: How you confirmed the infection of mice with S. uberis.

Comment3: Please mention the number of mice in each experimental and control group.

Comment4: Please confirm the RNA quality. The author needs to show the method for quality criteria.

Comment5: Please mention the % gel and amount of protein loaded in western blotting.

Comment6: 254-55- It confusing. The result does not present in the figure.

Comment7: 269-70-the text is not correct. UCP2 showed the difference between WT-B6+S. uberis and TLR2+ S. uberis  (Fig5A)

Comment8: SOD already decreased in the infected group compared to the WT. So TLR4 deletion may not be the reason for decrease SOD. Is my understanding correct?

Comment9: Fig5A and D- In mice, UCP2 increased after infection; however, in the cell line, UCP2 decreased following infection, please explain and discuss in the manuscript.

Comment10: 384-386- ECIST inhibition does not mean that TLR2 mediates mROS activation because TLR4 deleted mice also have reduced expression of ECIST (Fig 4B)

Comment11: Supplementary figures showed be cited in the text.

Comment12: 94 pleased begin the alphabetic, not 72, or the author may change the sentence.

Author Response

Thank you for reviewing our manuscript . Please see the attachment below.

Reviewer 2 Report

The file with revision is attached to this page

Author Response

Thank you for reviewing our manuscript. Please see the attachment below.

Round 2

Reviewer 1 Report

Thank you for your revised manuscript. The author improved all of the comments from me. This is now ready to accept. I am looking forward to seeing your new works in future. 

Reviewer 2 Report

Thank you